# Metabolic Signatures of 10 Processed and Non-processed Meat Products after In Vitro Digestion

**DOI:** 10.3390/metabo10070272

**Published:** 2020-07-03

**Authors:** Roland Wedekind, Pekka Keski-Rahkonen, Nivonirina Robinot, Frederic Mercier, Erwan Engel, Inge Huybrechts, Augustin Scalbert

**Affiliations:** 1Nutrition and Metabolism Section, International Agency for Research on Cancer, 69372 Lyon, France; wedekindr@students.iarc.fr (R.W.); KeskiP@iarc.fr (P.K.-R.); robinotn@iarc.fr (N.R.); HuybrechtsI@iarc.fr (I.H.); 2Micro-Contaminants, Aroma and Separation Sciences (MASS) Group, National Research Institute for Agriculture, Food and Environment (INRAE) UR370 QuaPA, 63122 Saint-Genès-Champanelle, France; frederic.mercier@inrae.fr (F.M.); erwan.engel@inrae.fr (E.E.)

**Keywords:** processed meat, smoked meat, fermented meat, pepper, syringol, biogenic amines, piperine, untargeted metabolomics, high resolution mass spectrometry

## Abstract

The intake of processed meat has been associated with several adverse health outcomes such as type II diabetes and cancer; however, the mechanisms are not fully understood. A better knowledge of the metabolite profiles of different processed and non-processed meat products from this heterogeneous food group could help in elucidating the mechanisms associated with these health effects. Thirty-three different commercial samples of ten processed and non-processed meat products were digested in triplicate with a standardized static in vitro digestion method in order to mimic profiles of small molecules formed in the gut upon digestion. A metabolomics approach based on high-resolution mass spectrometry was used to identify metabolite profiles specific to the various meat products. Processed meat products showed metabolite profiles clearly distinct from those of non-processed meat. Several discriminant features related to either specific ingredients or processing methods were identified. Those were, in particular, syringol compounds deposited in meat during smoking, biogenic amines formed during meat fermentation and piperine and related compounds characteristic of pepper used as an ingredient. These metabolites, characteristic of specific processed meat products, might be used as potential biomarkers of intake for these foods. They may also help in understanding the mechanisms linking processed meat intake and adverse health outcomes such as cancer.

## 1. Introduction

The intake of red and processed meat has been associated with an increased risk of several adverse health outcomes such as cancer [1], type II diabetes [2] and all-cause mortality [3]. Processed meat products form a diverse group of foods obtained from fresh meat using processing methods that include salting, curing, smoking, fermentation and drying to enhance shelf life and palatability. They can be categorized into fresh processed meat, cured meat pieces, raw-cooked products, precooked-cooked products, raw (dry)-fermented sausages and dried meat [4]. The products also vary in terms of pieces of meat used and the ingredients added such as curing salt, fat, spices and fillers.

Several mechanisms have been proposed to explain the carcinogenic effects of red and processed meats and the compounds involved in these mechanisms are often highly dependent on the cooking or processing methods applied to the meat products [5]. Some of these compounds can be found in both processed and non-processed meat such as carcinogenic polycyclic aromatic hydrocarbons (PAHs) which are mainly generated during the smoking or barbecuing of meat [6]. Other compounds are more specific to processed meat products. For instance, carcinogenic *N*-nitroso compounds are mainly due to the curing of meat with nitrite and nitrate [7]. The formation of heme-mediated carcinogenic lipid-oxidation by-products is increased by a high fat content of meat products, grinding and cooking of meat and exposure to oxygen [8,9], conditions which apply to many processed meat products.

Apart from the formation of process-induced toxicants, studies have also assessed the impact of meat processing on other outcomes such as protein digestibility [10], the evolution of the lipid profile [11] or flavor compounds [12]. Several studies have reported specific changes in compounds linked to different processing methods, such as the formation of guaiacol and syringol derivatives as aroma compounds in smoked meats [13,14] and that of biogenic amines during fermentation of dry-cured sausages [15,16]. 

The aforementioned studies have looked at a selection of compounds in a limited number of meat products. A better knowledge of variations in the composition of processed meat products could inform epidemiological studies elucidating the mechanisms by which intake of processed meat increases the risk of several adverse health outcomes. The aim of this study was to compare through an untargeted metabolomics approach the chemical profiles of 10 processed or non-processed meat product types consumed in Europe and to identify compounds specific to the various types of meat products or meat processing methods which might be useful as potential biomarkers of intake for these foods in epidemiological studies. This metabolomics approach was applied to in vitro digests of meat products in order to mimic profiles of small molecules formed in the gut upon digestion. The identification of metabolites characteristic of different types of meat products may shed new light on mechanisms linking intake of various meat products with risk of diseases.

## 2. Results

### 2.1. Metabolite Profiles of Meat Products

In vitro digests of 33 different meat samples from 10 processed meat types (Table 1) were analyzed by high resolution mass spectrometry coupled with ultra-high performance liquid chromatography (UHPLC–HRMS) and 5503 metabolite features present in at least four samples were selected for further analysis. When plotting the scores of the first and second principal components (PC1 and PC2) of a principal components analysis (PCA), it was found that meat products of the same type tend to cluster together (Figure 1). PC1 which accounts for 16% of the metabolite variations separates bacon from the other products. Some metabolite features that contribute to scores with negative loadings had exact masses matching with some lipids, and others with high positive loadings had exact masses matching with lipids and peptides in metabolite databases (Appendix A). The second principal component, which accounts for 13% of the variability, separates lean non-processed meat with high scores, cured meat pieces and fatty meat cuts with intermediate scores and different types of sausages with low scores. The features contributing with high absolute loading to PC2 are mainly lipophilic compounds that matched some lipids in databases (Appendix A).

### 2.2. Metabolites Significantly Elevated in Processed Meat Products

We further analyzed the same data to identify signals characteristic of processed meat products through univariate statistical analyses. There were 4581 metabolite features that had less than four missing values in at least one processed meat product category. We specifically looked for compounds significantly elevated in processed meat digests when compared to non-processed beef and pork digests that may result from meat processing. Out of those 4581 metabolite features, 178 were significantly elevated and 45 were significantly decreased in processed meat digests (Welch’s *t*-test, FDR-adjusted *p*-value < 0.05) with a fold change >two compared to non-processed meat digests (Appendix A). Only those metabolites showing a significant and large increase in processed meat, possibly be used as dietary biomarkers, were considered further. A heat map of the intensities of metabolite features elevated in processed meat digests is shown in Figure 2. In hierarchical clustering, six major clusters could be recognized. Cluster 6 includes syringol derivatives only found in smoked meat products; 4-propylsyringol and 4-allylsyringol are mainly found in hot dogs, whereas 4-methyl and 4-ethyl syringol are found in all smoked meats. Cluster 4 contains 14 compounds characteristic of black pepper. Their origin was established by comparison of their retention time and MS/MS fragmentation spectra with those of a black pepper extract (Appendix A). All these compounds with the exception of pepper compound 5 were also detected in extracts of white and green pepper (not shown). Figure 3 shows the intensity of six of the pepper compounds which could be identified at confidence level 1–2 in meat product digests [18,19,20,21]. All these pepper compounds show the highest levels in salami and fried sausage.

The dipeptide phenylalanyl-isoleucine was identified in metabolite cluster 2 primarily present in bacon. The remaining three clusters (clusters 2, 3 and 5) that show the highest intensities in sausage include mainly lipophilic, unknown compounds (unknown compounds 1–7, Appendix A). A summary of the metabolites that have been identified with confidence level 1 or 2 [22] are shown in Table 2.

The hierarchical clustering of the meat products reveals a similarity of all types of sausages and a clustering of non-processed meat products with cured meat pieces. Smoked ham samples (smoked products on the right, Figure 2) cluster together with other ham products rather than with smoked sausages or bacon.

### 2.3. Metabolites Specific for Fermented Meat Products

A separate analysis was performed to identify compounds that were significantly elevated in fermented products (*n* = 12) compared to non-processed pork (*n* = 18). A Welch’s t-test revealed several compounds that were elevated in fermented sausage and not found in all other processed meat products and non-processed meat. Among these are histamine, tryptamine and tyramine, which are all biogenic amines produced from amino acids by microbial metabolism (Table 3). In addition to these compounds, we specifically screened the data set for the presence of the biogenic amine γ-aminobutyric acid (GABA) previously reported in processed meat [15]. We extracted its intensities from the raw data and found it to be significantly elevated in fermented meat compared to non-processed pork. In non-fermented meat products, GABA, histamine and tryptamine were not detected and only trace amounts of tyramine were measured (Figure 4). 

## 3. Discussion

Processed meat products constitute a heterogeneous group of food products and their consumption may be associated with specific adverse health outcomes. However, the diversity of ingredients and processing methods such as curing, fermentation and smoking are rarely taken into account in epidemiological studies. 

The aim of this study was to explore global metabolite profiles of a large variety of processed meat products using an untargeted UHPLC-HRMS metabolomics approach. Cured meat pieces such as ham showed global metabolite profiles more similar to those of non-processed meat and clearly different from those of bacon and sausages. Score plots suggest that metabolite profiles were strongly influenced by some lipophilic metabolites in meat, with a gradient observed from lean meat cuts with lowest content to sausages with highest content, and cured meat pieces and fatty meat cuts with intermediate content.

The univariate comparison of metabolite profiles of processed and non-processed meat digests revealed more than a hundred metabolite features that were elevated in processed meat digests or only detected in these products. Clustering analysis revealed patterns characteristic of various types of processed meats. One cluster contained syringol derivatives only detected in smoked meats. We have previously published the identification of syringol metabolites in smoked meat and that of their sulfated metabolites in human urine and plasma and showed that they could be used as biomarkers for smoked meat intake [23].

A second cluster of metabolites which showed high intensity in sausages contained several piperamides from black pepper. Pepper is an important ingredient of many processed meat products used for its flavor, but also for its antioxidative and antimicrobial properties [24,25]. Volatile terpenes derived from pepper were previously characterized as aroma compounds of dry-cured sausages [12,26]. However, we are not aware of any study reporting non-volatile pepper compounds such as the piperamides in processed meat products. We could identify fourteen metabolites derived from black pepper that discriminate processed from non-processed meat. These metabolites showed very high intensities in the sausage digests compared to non-processed meat digests. These compounds increase shelf life of processed meat products, and may also exert some beneficial effects on human health as suggested by their in vitro biological properties [27]. More particularly, piperine was shown to have cytotoxic effects on cancer cells [27] and its blood levels have been associated with lower risk of breast cancer in epidemiological studies [28,29]. On the other hand, piperine levels in blood were found to be associated with an increased risk of diabetes [30]. In order to evaluate if the pepper compounds in processed meat products might have an effect on human health, further studies are needed to estimate whether processed meat products contribute a significant proportion to the total dietary pepper exposure.

The comparison of metabolite profiles of fermented sausages and non-processed pork showed higher levels of the biogenic amines tyramine, histamine, tryptamine and γ-aminobutyric acid (GABA) in the fermented products. This is in line with the known microbial decarboxylation of amino acids during meat fermentation [15,16]. Microbial activity of fermented meat products leads to the desired flavor and enhanced shelf life of the products, but high amounts of some biogenic amines such as histamine can be toxic to the consumer [16]. GABA is not only produced by meat fermentation. It is also naturally present in some vegetables and cereals and is increased in fermented foods such as dairy products and soy sauce [31]. It has been shown to have beneficial effects on blood pressure [31]. Biogenic amines may also be present in unintentionally fermented meat products, but in our study no product that does not include fermentation in its processing methods showed increased levels of these metabolites. 

This study has a few limitations. The method employed is not universal in its ability to detect all possibly present compounds, and thus some of the expected metabolites known to discriminate processed and non-processed meats were not found. Syringol compounds are detected in this work in contrast to the structurally related guaiacol compounds that have been shown to be present in smoked meat [14]. The aim of this study was to explore the general metabolite profiles of meat products using an established analytical procedure, and data on process-induced toxicants for specific meat products can be found elsewhere [6,7,32]. Another limitation that is common with most untargeted metabolomics studies is the low number of compounds that could be annotated with high certainty. In our study, several compounds that showed high loadings in the PCA were tentatively annotated as lipids, small peptides and pepper-derived compounds, but their exact identity still remains unknown. However, the comparison of MS/MS fragmentation spectra with those of the compounds from the pepper extract gives us high confidence in the specific origin of a number of compounds derived from this specific ingredient. Using an in vitro approach, we were able to identify several compounds that are specific for different processing methods and that might be used as biomarkers of intake for these foods. However, the analysis of human blood and urine samples from dietary intervention and observational studies is needed to assess if these compounds or their metabolites can be detected in human biofluids and if they predict intake of these products.

This study also has several strengths, the main one being the application of an untargeted metabolomics approach which allowed for the detection of thousands of metabolite features in meat and the identification of compounds able to discriminate specific types of meat products. This allowed for the discovery of pepper-derived compounds as some of the main discriminants for processed meats. Another strength of this work is the large variety of meat products studied. We are not aware of other metabolomics studies comparing such a diverse set of meat in vitro digests. 

## 4. Materials and Methods 

### 4.1. Reagents and Meat Products

Analytical standards were purchased from Sigma, France. At least three different varieties of each meat product (pork, lean and fatty cuts; beef, lean and fatty cuts; hot dogs; salami type sausage; raw ham; cooked ham; raw sausages to be fried; bacon) were purchased in local supermarkets and butcher’s shops in Lyon, France (Table 1). Meat products were chosen to represent different processing methods (curing, fermentation, and smoking) and to cover the major processed meat products consumed in many European countries [33]. Whole pepper corns (white, black, or green; Ducros, Avignon, France) were purchased in a local supermarket. 

### 4.2. In Vitro Digestion

Meat products were first fried in a non-stick pan (no oil added) if commonly consumed heated (pork, beef, fried sausage, bacon and hot dogs). The in vitro digestion of the processed meat products is described in detail elsewhere [23]. All products were minced to a chunky homogenate using a blender and 3 aliquots of 1.5 g were digested following a standardized protocol [34]. The concentration of electrolytes in the buffer are based on human in vivo data and can be found in the original publication [34]. In a salivary phase, each aliquot homogenate was incubated with 1.5 mL of buffer for 2 min (pH = 7, no amylase was used because of the negligible amount of starch in the products). The same homogenate was then treated in the gastric phase with 3 mL of buffer (pH = 3) containing pepsin for 120 min. Finally, in the intestinal phase, 6 mL of buffer containing pancreatin and bile extracts was added, the pH increased to 7 and the mixture shaken for 120 min. All steps were performed at 37 °C. Digests were finally centrifuged (15 min, 15,000× *g*, 22 °C) and the supernatants stored at −80 °C until analysis. 

### 4.3. Liquid Chromatography-Mass Spectrometry Analysis

Supernatants of in vitro digests (30 µL) were mixed with cold acetonitrile (200 µL) and centrifuged (3220× *g*, 10 min, 4 °C). The supernatant was filtered on 0.2 μm Captiva ND plates (Agilent Technologies, Santa Clara, CA, USA) and filtrates diluted 10 times with acetonitrile. Commercial pepper (white, black, green) was ground with a mortar and pestle, and 9.5 mg were added to 9:1 acetonitrile:water (vol:vol; 10 mL), shaken for 60 min, filtered on 0.2 μm Captiva ND plates, diluted 100 times with the same solvent, and stored at −80 °C. Pepper extracts were diluted 10 times with acetonitrile before analysis.

Sample extracts were analysed by LC-MS using a 1290 Binary LC system coupled with a 6550 quadrupole time-of-flight (QTOF) mass spectrometer equipped with a jet steam electrospray ionization source (Agilent Technologies, Santa Clara, CA, USA), as previously described [35]. In short, samples were ordered randomly in the batch. A quality control (QC) sample consisting of a pool of all samples in one batch was analysed for every eight study samples injected. Two microliters of sample extracts were injected onto a reversed phase C18 column (ACQUITY UPLC HSS T3 2.1 × 100 mm, 1.8 μm, Waters, Saint-Quentin-En-Yvelines, France) maintained at 45 °C. A linear gradient made of ultrapure water and LC-MS grade methanol, both containing 0.05% (*v/v*) of formic acid, was used for elution at a flow rate of 0.4 mL/min: 0–6 min: 5–100% methanol, 6–10.5 min: 100% methanol, 10.5–13 min: 5% methanol. The mass spectrometer was operated in positive ionization mode, detecting ions across a mass range of 50–1000 daltons.

### 4.4. Data Processing

The raw data were processed using the MassHunter software package (Mass Profiler Professional version B14.9.1, Qualitative Analysis version B06.00 and DA Reprocessor version B.05.00; Agilent technologies, Santa Clara, CA, USA) with Agilent’s recursive feature finding method. For this, metabolite features were extracted from the raw data and aligned over all samples of the experiment based on mass (+/− 15 ppm) and retention time (+/− 0.1 min). In a second, targeted recursive search, all features that were present in more than 3 samples in the experiment were extracted from the raw data. The resulting data were aligned with the same criteria to obtain the final feature table. If no signal was found for a metabolite feature in a sample, the intensity was imputed as 1. Detailed parameters can be found in Appendix A.

### 4.5. Metabolite Annotation

Metabolite features were annotated by searching metabolites with matching mass in the Human Metabolome Database (HMDB) and METLIN as [M+H]^+^ and [M+Na]^+^ ions with a mass tolerance of 8 ppm [36,37]. For the compounds specific of meat processing, targeted fragmentation (MS/MS) spectra were acquired at 10 and 20 eV and compared to those of commercially available standards, in-house synthesised standards or to compounds extracted from liquid smoke or black pepper. Levels of confidence given to each annotated peak were those defined by Sumner et al. [22] where level 1 corresponds to matching exact mass, retention time and MS/MS fragmentation spectrum with those of an authentic chemical standard. Level 2 corresponds to very probable annotation based on matching exact mass, similar physico-chemical properties and MS/MS spectrum with those curated in HMDB and METLIN or the references cited in Appendix A. Annotation evidence is given in Appendix A.

### 4.6. Statistical Analysis

For principal component analysis (PCA), only metabolite features with an intensity > 1 in at least 4 out of the 99 samples were included and the data were log2-transformed. For the univariate analyses, only metabolite features that had less than 4 missing values in at least one processed meat product category (e.g., out of 9 hot dog samples or 12 raw ham samples, etc.) were included in the analysis of the in vitro digests. Univariate analysis was carried out with log2-transformed data. Non-paired Welch’s *t*-tests were used to identify metabolite features that were significantly elevated in in vitro digests of processed meat products compared to non-processed meat. The results were adjusted for multiple testing using the Benjamini–Hochberg method and a false discovery rate of 0.05. Only compounds that showed at least a twofold increase in the intensity of processed meat compared to non-processed meat after log transformations were retained for annotation. A second Welch’s *t*-test was performed to identify all features that were significantly elevated in salami digests compared to digests of lean and fatty pork. The biogenic amine γ-aminobutyric acid (GABA) has been reported to be a microbial metabolite in fermented meat but was not included in the first feature table. Its intensities were manually extracted from the raw data using the Agilent Profinder software (+/− 8 ppm) and added to the feature table for the comparison of fermented sausage vs. pork. All statistical analyses and graphical presentations were carried out using the R open-source statistical computing software version 3.6.1 (R Foundation for Statistical Computing, Vienna, Austria). The heat map image was prepared using the *ComplexHeatmap* package in R [38], with Euclidean distance for hierarchical clustering.

## 5. Conclusions

In this study, we were able to explore the unique metabolite profiles of diverse processed and non-processed meat products after an in vitro digestion. Metabolite profiles of cured meat pieces such as ham were more similar to non-processed meat than to sausages or bacon. A wide variety of metabolites were significantly elevated in processed meat products compared to the non-processed meats, and metabolite profiles of meat products depended on their ingredients and processing methods. Syringol derivatives and biogenic amines were identified as specific markers for smoked and fermented processed meats, respectively. We could identify several natural compounds originating from pepper in processed meats, with the highest levels found in sausages.

## Figures and Tables

**Figure 1 metabolites-10-00272-f001:**
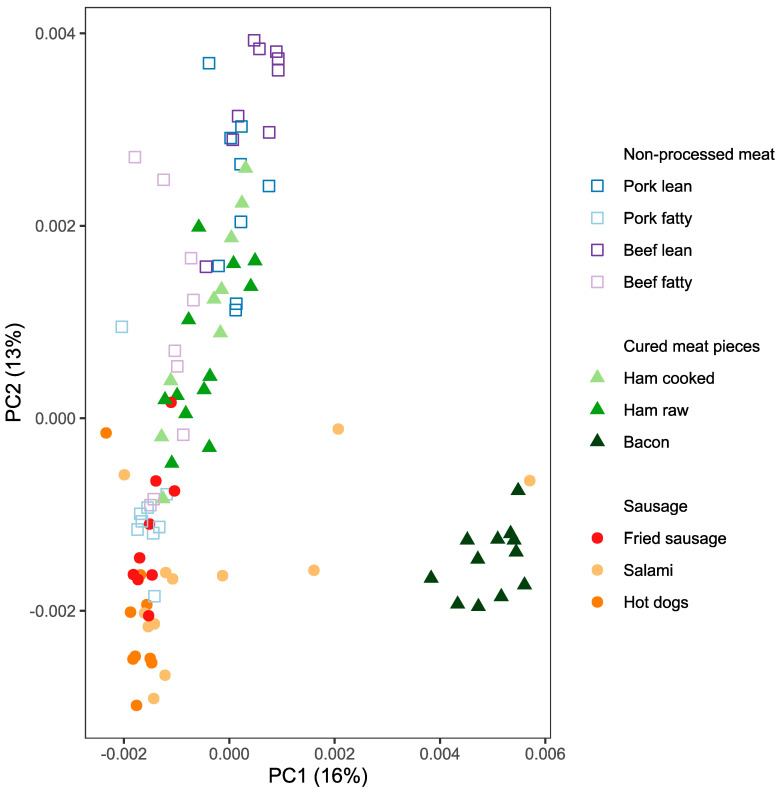
Principal component analysis of 5503 metabolite features detected in meat in vitro digests. Thirty-three meat products were digested in triplicate and the bioavailable fractions were analyzed by untargeted UHPLC–HRMS-based metabolomics. The figure shows the scores plot of the first and the second principal component which represent 16% and 13% of the metabolites’ variability, respectively. Metabolite intensities were normalized by total ion count of each sample to account for differences in water content between products.

**Figure 2 metabolites-10-00272-f002:**
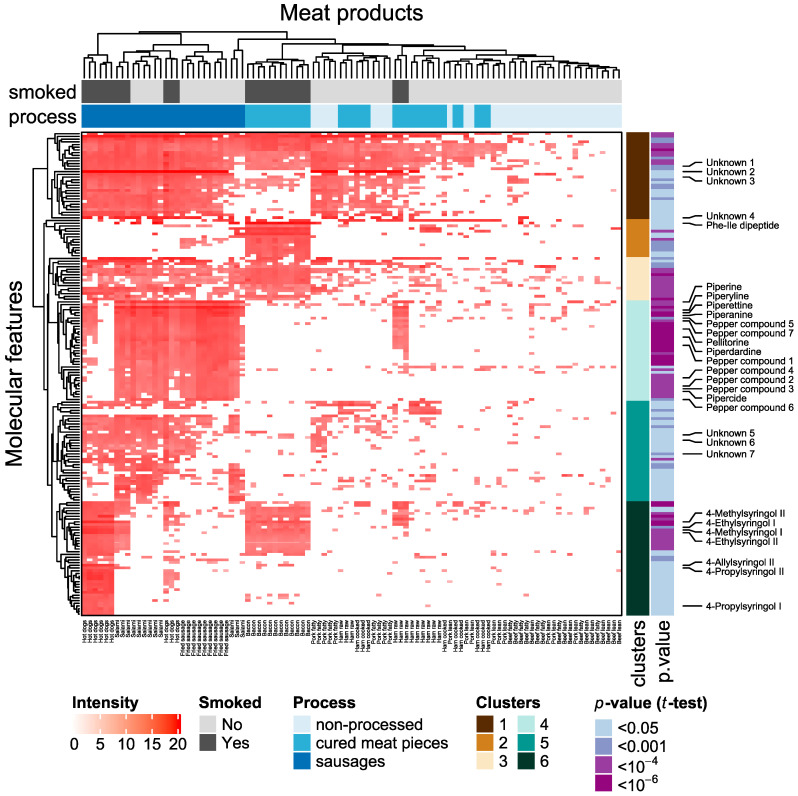
Heat map showing intensities of metabolite features with significantly higher levels in processed meat digests compared to non-processed meat digests. A Welch’s t-test revealed 178 metabolite features with intensities significantly (FDR-adjusted *p*-value < 0.05) and increased at least twofold in processed meat products (*n* = 63) compared to non-processed red meat (*n* = 36). Intensities were log transformed. Meat products are displayed in columns and colored by the product type. Metabolite features (rows) are colored by the FDR-adjusted *p*-value of the *t*-test. Names of identified metabolites are indicated. Details on these compounds with clusters they belong to are given in Table 2 and Appendix A
Appendix A.

**Figure 3 metabolites-10-00272-f003:**
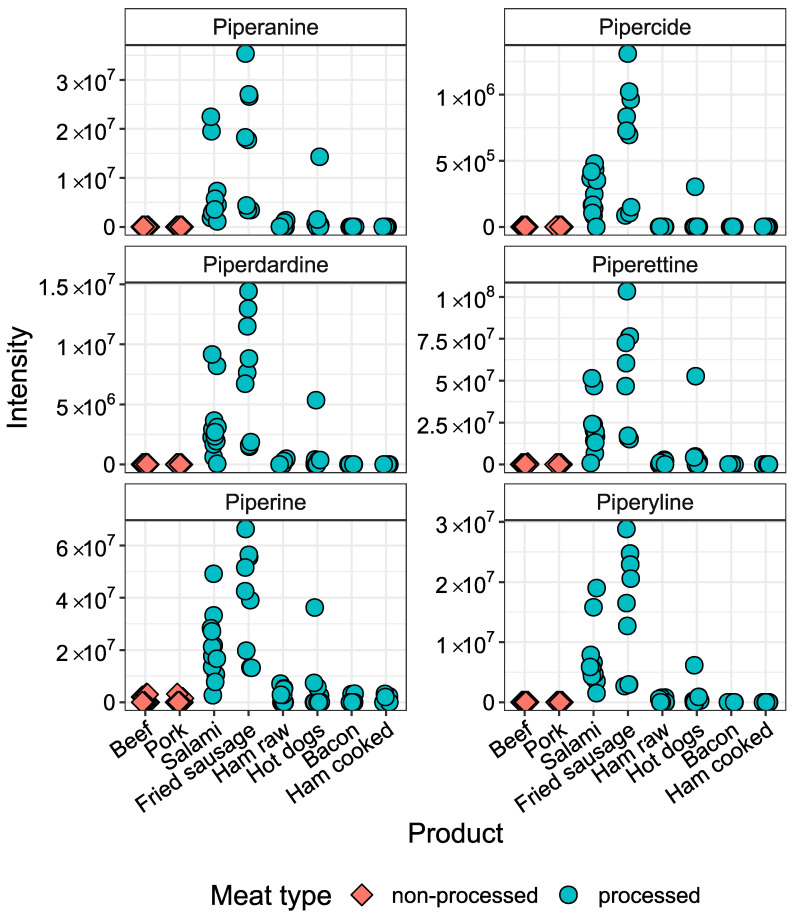
Intensities of pepper metabolites identified in in vitro digests of meat products. All compounds are significantly elevated (FDR-adjusted *p*-value < 0.05) in processed meat products (*n* = 63) compared to non-processed red meat products (*n* = 36).

**Figure 4 metabolites-10-00272-f004:**
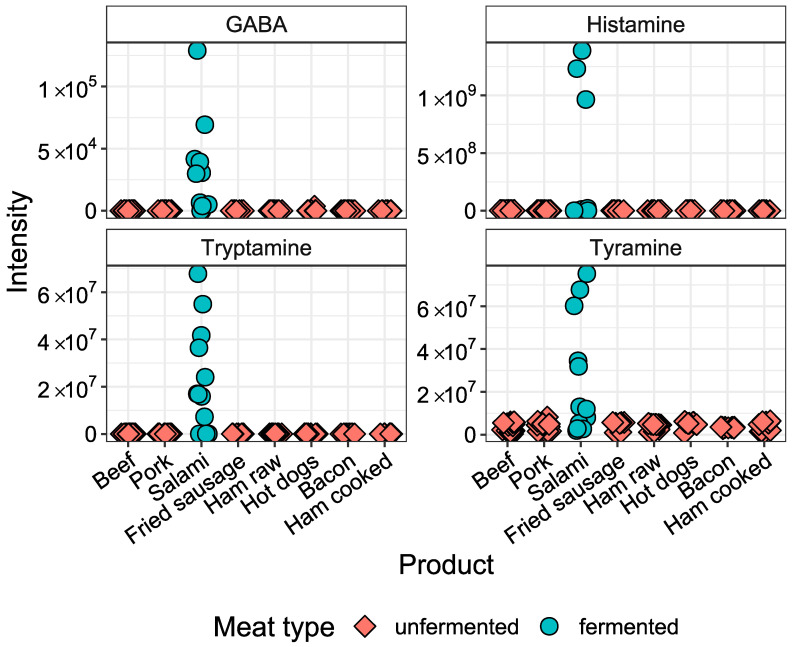
Intensities of biogenic amines in in vitro digests of meat products. Shown are four biogenic amines which showed significantly higher levels (FDR-corrected *p*-value < 0.05) in fermented sausages (*n* = 12) compared to non-processed pork (*n* = 18).

**Table 1 metabolites-10-00272-t001:** Characteristics of meat products purchased in local supermarkets and butcher shops and digested in vitro in triplicate.

Product	Product Type	Number of Samples	Number of Processed Products ^e^	Number of Smoked Products	Number of Fermented Products	Number of Products from		Fat Content (g/100g Product) ^f^	Protein Content (g/100g Product) ^f^
						Butcher	Supermarket		
Lean pork ^a^	Non-processed	3	0	0	0	3	0	2.4	21.9
Fatty pork ^b^	Non-processed	3	0	0	0	3	0	10	20.3
Lean beef ^c^	Non-processed	3	0	0	0	3	0	4.2	21.5
Fatty beef ^d^	Non-processed	3	0	0	0	3	0	6.6	22.6
Cooked ham	Cured meat pieces	3	3	0	0	1	2	3.0 ± 0.7	19.8 ± 1.3
Raw ham	Cured meat pieces	4	4	1	0	0	4	13.0 ± 3.5	28.8 ± 2.2
Bacon	Cured meat pieces	4	4	4	0	0	4	23.4 ± 3	16.8 ± 0.9
Fried sausage	Sausage	3	3	0	0	1	2	16.2 ± 9.3	15.5 ± 2.8
Salami, Dry-cured sausage	Sausage	4	4	1	4	0	4	32.8 ± 8.3	24.5 ± 5.1
Hot dogs, frankfurter sausage	Sausage	3	3	3	0	0	3	2.5.3± 2.5	12.5 ± 0.5

^a^ Pork tenderloin. ^b^ Pork neck. ^c^ Filet mignon (beef), lean cut of entrecote. ^d^ Entrecote, flank steak. ^e^ All processed meats are salted and cured ^f^ The contents of fat and protein were calculated based on nutrition information on packaging for products purchased from supermarkets and based on the German Nutrient Data Base [17] for products purchased from butchers.

**Table 2 metabolites-10-00272-t002:** Selected compounds significantly elevated in digests of processed meat compared to digests of non-processed meat.

Compound	PubChem CID	Measured *m/z*	RT (min)	Adjusted *p*-Value ^a^	Metabolite Cluster ^b^	Level of Confidence ^c^	Mass Accuracy (ppm)
Piperine	638024	308.1262 ^d^	6.01	3.7 × 10^−5^	4	1	2
Piperyline	636537	272.1287	5.74	7.4 × 10^−5^	4	2	2
Piperettine	101878852	312.1597	6.29	6.9 × 10^−7^	4	2	1
Piperanine	5320618	288.1601	5.93	8.0 × 10^−8^	4	2	2
Piperdardine	10086948	314.1751	6.28	1.6 × 10^−7^	4	2	0
Pipercide	5372162	356.222	6.70	9.3 × 10^−6^	4	2	0
Pellitorine	5318516	224.2012	6.34	4.8 × 10^−7^	4	1	1
4-Ethylsyringol	61712	183.1018 205.0841 ^d^	5.08	0.00011	6	1	1
4-Methylsyringol	240925	169.086 191.0686 ^d^	4.57	5.3 × 10^−6^	6	1	0
4-Allylsyringol	5352905	195.102 217.0846 ^d^	5.32	0.019	6	1	2
4-propylsyringol	524975	197.1173 219.1001 ^d^	5.54	0.020	6	1	0
Phenylalanyl-Isoleucine	7010566	279.1712	3.36	0.016	2	2	3

Abbreviations: retention time (RT); parts per million (ppm).^a^
*p*-Values of Welch’s *t*-test comparing intensities of metabolites in processed meat digests vs. non-processed meat digests adjusted for multiple testing using the FDR-method. ^b^ Metabolite clusters correspond to the hierarchical clustering shown in the heat map of Figure 2. ^c^ Level 1 identification corresponds to matching exact mass, retention time and MS/MS fragmentation pattern with an authentic chemical standard, level 2 corresponds to matching exact mass and MS/MS fragmentation pattern to spectral databases [22]. ^d^ [M+Na]^+^ ions; all other ions are [M+H]^+^.

**Table 3 metabolites-10-00272-t003:** Biogenic amines significantly elevated in digests of fermented sausages compared to digests of non-processed pork. All compounds were identified by comparison of exact mass, retention time and MS/MS fragmentation pattern to those of chemical standards.

Compound	*m/z*	RT ^a^ (min)	RT Standard (min)	Adjusted *p*-Value ^b^	Mass Accuracy ^c^ (ppm)	PubChem CID	Level of Confidence ^d^
GABA(γ-Aminobutyric acid)	104.0709	0.61	0.59	0.0025	4	119	1
Histamine	112.0869	0.50	0.48	0.019	0	774	1
Tyramine	138.0914	1.31	1.31	0.016	2	5610	1
Tryptamine	161.1076	2.39	2.36	0.0025	2	1150	1

^a^ Retention time (RT); ^b^ FDR-adjusted *p*-value of a welch *t*-test comparing metabolite intensities in fermented sausage digests to non-processed pork digests; ^c^ accuracy of measured mass compared to the theoretical mass of the annotation; ^d^ annotation level of confidence: identity confirmed by comparison of RT and MS/MS fragmentation pattern with chemical standard (level 1).

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
