# Peer review of "Metabolic Signatures of 10 Processed and Non-processed Meat Products after In Vitro Digestion"

_metabolites, 2020, doi:10.3390/metabo10070272_

Round 1
Reviewer 1 Report
Wedekind et al examined the metabolic signatures of 33 processed and non-processed meat products after in vitro digestion. The authors found that processed meat products showed metabolites profiles that were clearly distinct from those of non-processed meat products. The authors found that syringol compounds were increased in meat during smoking and biogenic amines were formed during meat fermentation. The authors also found piperine and related pepper compounds as well. This is an important analysis, but I have a few suggestions/questions for consideration:
- I was unable to view Supp Table 1 as it was converted to a PDF and could not view the full table.
- Figure 2: The authors only present metabolite features that were 2-fold increased in the processed vs unprocessed meat products. It would be also useful to see if metabolites were higher (if any) in the unprocessed meat vs processed meat categories (fold decrease)
- Line 292: data ‘were’ NOT data “are”
- What was the rationale for only choosing a ≥2-fold increase?
- An important limitation of this study is that these metabolites were discovered using in vitro digests and not from human biospecimens such as plasma or urine. This should be acknowledged in the Discussion.
- Supp Table 3: Rename title as “Features significantly elevated in processed meat products compared to non-processed meat products….”
Author Response
Reviewer 1:
- I was unable to view Supp Table 1 as it was converted to a PDF and could not view the full table.
Reply : We are sorry for that. It is certainly difficult to convert such an Excel file into a pdf file.
- Figure 2: The authors only present metabolite features that were 2-fold increased in the processed vs unprocessed meat products. It would be also useful to see if metabolites were higher (if any) in the unprocessed meat vs processed meat categories (fold decrease)
Reply: Fourty-five metabolite features were at least 2-fold elevated in non-processed meat compared to processed meat. We have added this information to the results section (lines 98-102).
- Line 292: data ‘were’ NOT data “are”
Reply: We could not find “data are” in the text. It might have been changed by the editors already?
- What was the rationale for only choosing a ≥2-fold increase?
Reply: In this study we aimed at identifying compounds specific for meat processing. We chose an arbitrary fold-change threshold that seemed high enough to only retain compounds that were strongly different between the various meat types. Lower thresholds such as a fold-change increase of 1.25 might have led to the inclusion of many metabolites related to content of fat and muscle tissue which was not the focus of this study. A sentence has been added (lines 100-101).
- An important limitation of this study is that these metabolites were discovered using in vitro digests and not from human biospecimens such as plasma or urine. This should be acknowledged in the Discussion.
Reply: We thank the reviewer for this comment. This study on meat digests was a first step to help in discovering biomarkers of processed meat intake. We do not think it is an intrinsic limitation of this study. A dietary intervention study and an observational study were subsequently conducted (see cited reference #23). A few lines (lines 224-228) were added to clarify this point.
- Supp Table 3: Rename title as “Features significantly elevated in processed meat products compared to non-processed meat products….”
Reply: We have made the changes accordingly.
Reviewer 2 Report
The manuscript by Wedekind et al. presents a study where the in vitro digestion of 33 different processed meat products (representing 10 different types of product categories) are investigated. The investigation consists of an LC-MS-based metabolomics analysis of the in vitro digests. As such, the study is rather simple as it only includes LC-MS-based metabolomics analyses. However, the work appears sound and solid, and the LC-MS compound assignments reported are an important contribution to this research field. The manuscript also appears very clear and well written, and I only have minor comments provided below.
Title and elsewhere: The study includes analysis of samples from a total of 33 different (commercial) meat products. Nevertheless, of these, some products are representing replicates of same product type, and it would be more clear to refer to the number of product types (which appears to be 10?) than the total number of meat products included. Thus, revision of the paper title is recommended.
The study identifies a number of metabolites that appears specific for fermented meat products. One of these is γ-aminobutyric acid (GABA). A vast amount of studies report that GABA is a health-promoting functional compound. This fact is not mentioned in the manuscript, and in the discussion section, the possible impact of GABA should be described more nuanced to highlight that it may not necessarily be a ‘negative’ compound. (GABA is also found in tomato, which may also be mentioned).
Author Response
The manuscript by Wedekind et al. presents a study where the in vitro digestion of 33 different processed meat products (representing 10 different types of product categories) are investigated. The investigation consists of an LC-MS-based metabolomics analysis of the in vitro digests. As such, the study is rather simple as it only includes LC-MS-based metabolomics analyses. However, the work appears sound and solid, and the LC-MS compound assignments reported are an important contribution to this research field. The manuscript also appears very clear and well written, and I only have minor comments provided below.
Title and elsewhere: The study includes analysis of samples from a total of 33 different (commercial) meat products. Nevertheless, of these, some products are representing replicates of same product type, and it would be more clear to refer to the number of product types (which appears to be 10?) than the total number of meat products included. Thus, revision of the paper title is recommended.
Reply: We agree with the reviewer. Speaking of 33 different products might be misinterpreted. We have followed the suggestion of the reviewer and changed it to 10 product types (lines 2, 15, 58-59, 68)
The study identifies a number of metabolites that appears specific for fermented meat products. One of these is γ-aminobutyric acid (GABA). A vast amount of studies report that GABA is a health-promoting functional compound. This fact is not mentioned in the manuscript, and in the discussion section, the possible impact of GABA should be described more nuanced to highlight that it may not necessarily be a ‘negative’ compound. (GABA is also found in tomato, which may also be mentioned).
Reply: We have changed the discussion accordingly (Lines 205-209) to show that biogenic amines have a more complex impact on health with some toxic (e.g. histamine) and some beneficial effects (e.g. GABA).